# Comparative Analysis of Saccharification Characteristics of Different Type Sweetpotato Cultivars

**DOI:** 10.3390/foods12203785

**Published:** 2023-10-15

**Authors:** Chen Li, Meng Kou, Weihan Song, Mohamed Hamed Arisha, Runfei Gao, Wei Tang, Hui Yan, Xin Wang, Yungang Zhang, Qiang Li

**Affiliations:** 1Xuzhou Institute of Agricultural Sciences in Jiangsu Xuhuai District/Sweetpotato Research Institute, Chinese Academy of Agricultural Sciences/Key Laboratory of Biology and Genetic Breeding of Sweetpotato, Ministry of Agriculture and Rural Affairs, Xuzhou 221131, China; lichen_xz@163.com (C.L.); koumeng2113@163.com (M.K.); xzsongweihan@163.com (W.S.); 15852141027@163.com (R.G.); tangweilhr@gmail.com (W.T.); yanhui_sweetpotato@163.com (H.Y.); wangxin@jaas.ac.cn (X.W.); zhangyungang@jaas.ac.cn (Y.Z.); 2Department of Horticulture, Faculty of Agriculture, Zagazig University, Zagazig 44511, Egypt; mohhamedarisha@gmail.com

**Keywords:** *Ipomoea batata*, saccharification, maltose, starch, gelatinization temperature

## Abstract

As an important characteristic crop in China, sweetpotato plays an important role in the intake and supplement of nutrients. The saccharification characteristics of sweetpotato determine the edible quality and processing type. Exploring the saccharification characteristics of sweetpotato is of great significance to the selection of processing materials and the formation mechanism of service quality, but there are few relevant studies. A comparison study of two high saccharification varieties (Y25 and Z13) and one low saccharification variety (X27) was conducted to analyze their storage roots physical and chemical properties. The results show that the dry matter content, starch, and amylose content of Y25 and Z13 were significantly different from those of X27. Furthermore, the total amylase activity was significantly higher than that of X27. On the other hand, the starch gelatinization temperature was significantly lower than that of X27. The starch reduction in Y25 and Z13 is four times more than that in X27, and the maltose content of Y25 and Z13 is more than two times that of X27. Finally, the scores of sensory evaluation and physiological sweetness were significantly higher than those of X27. The results provide a theoretical basis for understanding the saccharification characteristics of sweetpotato varieties and are of guiding significance for the selection of sweetpotato parents.

## 1. Introduction

Sweetpotato (*Ipomoea batatas* (L.) Lam.) is an important food and industrial raw material crop in China which plays an important role in the national economy. In addition, sweetpotatoes, which are rich in carbohydrates and antioxidants, are environmentally friendly crops and bottom-line crops to ensure food security and are widely planted in countries all over the world [1,2,3]. The annual output of sweetpotatoes exceeds 90 million tons, of which Asia accounts for about 71.1 percent, with China making the largest contribution [4].

The saccharification characteristic of sweetpotato is an important index for evaluating the edible quality and processing quality. Fresh sweetpotatoes require high saccharification varieties which can produce more soluble sugars and improve the overall taste. In addition, processing roasted sweetpotatoes, slices, and preserves all require highly saccharified sweetpotato varieties, while starch processing requires sweetpotato varieties with low saccharification efficiency to reduce starch decomposition and maintain higher starch content. The saccharification efficiency of sweetpotato depends not only on the genotypic differences of varieties [5] but also on the growth environment [6] and storage conditions [7] of sweetpotato. Therefore, the saccharification characteristics of sweet potato determine its use [8]. 

The saccharification of sweetpotato is mainly the process of starch decomposing into soluble sugars including maltose, sucrose, fructose, and glucose in storage root. In the fresh sweetpotato storage roots, soluble sugar mainly includes sucrose, fructose, glucose, and a very small amount of maltose, but with the ripening process of sweetpotato, starch decomposes to produce a large amount of maltose, while sucrose, fructose, and glucose change to become present in smaller percentages [9,10]. Therefore, sweetpotato saccharification is mainly the process of starch decomposition to maltose. In the ripening process of sweetpotato, the synthesis of maltose belongs to the metabolic pathway of starch and sucrose, which involves many complex physiological and biochemical reactions. Until now, the molecular mechanism and the regulatory pathway of maltose synthesis are still not clear. Related studies suggest that saccharification-related enzymes (especially α-amylase and β-amylase) play a promoting role in the ripening process of sweetpotato, decomposing starch to synthesize a large amount of maltose [11]. Through the study of sweetpotato varieties bred in Japan, it was found that the activity of amylase was positively correlated with the content of maltose, but it would not increase when the concentration of maltose reached a certain concentration [12,13,14]. At the same time, it was found that the content of maltose was positively correlated with the sweetness of sweetpotato [15]. The above results show that in addition to the effect of amylase on the formation of maltose, there are many factors that promote or restrict the formation of maltose. Another factor may be related to starch gelatinization, because amylase cannot decompose raw starch granules, and starch gelatinization provides amylase with the necessary reaction substrate [16]. Further study found that there was a significant negative correlation between maltose concentration and sweetpotato starch gelatinization temperature, and the negative correlation was more obvious when the amylase activity was stronger [15]. It is not clear whether there are other factors affecting sweetpotato saccharification, and there are few reports on the systematic study of the sweetpotato saccharification process.

Therefore, studying the saccharification characteristics of sweetpotatoes has important practical significance for industrial utilization and material selection. In this study, three sweetpotato varieties with different saccharification were selected, their physical and chemical properties were compared, and their saccharification characteristics were analyzed in order to provide materials and theoretical basis for the follow-up study of the sweetpotato saccharification molecular regulation mechanism.

## 2. Materials and Methods

### 2.1. Material

During daily processing, we found significant differences in sweetness and sugar content among Y25, Z13, and X27. Therefore, we selected these three varieties as materials to study the differences in saccharification characteristics among different sweetpotato varieties. Xushu 27 (X27), with red skin and white flesh, has low saccharification efficiency and low sweetness after cooking, which was bred by Xuzhou Institute of Agricultural Sciences in Jiangsu Xuhuai District. Yanshu 25 (Y25) has high saccharification efficiency and sweetness, which is an edible sweetpotato widely grown in China, with light red skin and orange flesh. Zheshu13 (Z13) is an important processed variety with high sweetness, and its skin is red and flesh color is yellow.

### 2.2. Experimental Design

The experimental materials were planted in the modern agricultural experimental demonstration base of Xuzhou Institute of Agricultural Sciences in Jiangsu Xuhuai District. The three sweetpotato varieties were arranged in random blocks and planted in single ridges with plant spacing of 25 cm and row spacing of 85 cm. Two growth periods were set for sample collection, which were 90 d (planting time of early harvest sweetpotato) and 130 d (planting time of normal harvest sweetpotato) after transplanting. 10 plants were randomly harvested from each variety with 3 replicates and a total of 30 plants were collected. Medium-sized sweetpotato storage roots were selected for follow-up experiments.

Fresh sweetpotato storage roots were treated with CK (fresh sweet potato was not treated), treated at 60 °C, 80 °C and 90 °C for 1.5 h, and steamed. In order to investigate the change in the sugar content of sweetpotato storage roots during heating, the above temperatures were set. At the same time, considering that the saccharification mainly occurs in the high temperature stage, the temperature treatment was mainly concentrated in the range of 60–100 °C. After peeling, the middle part was taken, soaked in liquid nitrogen, and transferred to the ultra-low temperature refrigerator at −80 °C until the analysis was completed. The samples for cytological analysis were fixed in a FAA fixation solution at 4 °C for 24 h and paraffin sections were prepared.

### 2.3. Physicochemical Index Determination

#### 2.3.1. Determination of Dry Matter Content, Starch Content, Amylose Content, and Amylase Activity

The dry matter content was determined by the freeze-drying method. Starch content was determined by using the starch content assay kit (Hefei Laier Biological Technology Co., Hefei, China) and amylose content was determined using the amylose assay kit (Megazyme, Bray, Ireland), according to the instructions.

#### 2.3.2. Analysis of Gelatinization Characteristics of Starch

Sweetpotato starch was extracted by the water washing precipitation method [17]. We accurately weighed 3.0 g sweetpotato starch into a RVA gelatinizing box and added 25 mL ddH_2_O to make the concentration of starch solution (10.7%). The RVA program setting was as follows: RVA rotor at 960 r/min speed of 10 s and then at 160 r/min speed until the end of the experiment. The initial temperature was 50 °C for 1 min, then it rose to 95 °C for 4.5 min after 4 min, and then dropped to 50 °C for 3 min after 4 min; #the whole process lasted 16.7 min. Data were analyzed with RVA data analysis software (Thermocline for Windows (TCW3)).

#### 2.3.3. Sensory Evaluation, Physiological Sweetness, and Electronic Tongue Analysis of Sweetpotato

Sensory evaluation was based on the method as Zhang et al. described [18]. Five trained evaluators tasted the steamed sweetpotato, scored the sweetness, fiber content, taste, comprehensive score, etc., and averaged the final score.

The biochemical sweetness of each sweetpotato was calculated according to the literature method [10]. Physiological sweetness = sucrose content × 1.0 + fructose content × 0.7 + glucose content × 0.5 + maltose content × 0.25.

Electronic tongue sensory evaluation sample treatment and determination condition setting was as follows: a weight of 40 g ripe sweetpotato sample was taken with an addition of 100 mL ddH_2_O to homogenize the pulp in a cooking machine, and the liquid was transferred to a centrifugal tube. After adding 100 mL ddH_2_O several times, the washing liquid was collected and centrifuged in a centrifugal tube at 3000 rpm/min for 10 min. After filtration, the supernatant wa pumped and filtered, and the filtrate was stored in a refrigerator at −20 °C for machine analysis. The filtrate was placed in the sample cup and then placed in the electronic tongue sample automatic sampler. The five-taste sensor was set for four times and the sweetness sensor was determined for five times; the measurement data were automatically collected. We used the Taste Analysis data analysis software (Insent Intelligent Sensor Technology, Inc., Kanagawa-ken, Japan) for data transformation and analysis.

#### 2.3.4. Observation on Tissue Structure of Storage Root in Sweetpotato

For the preparation method of paraffin slices of sweetpotato, we referred to the method of Soukup et al. [19], which was slightly modified on the original basis. The samples treated at different temperatures were fixed in a FAA fixed solution (38% formaldehyde 5 mL; glacial acetic acid 5 mL/70% alcohol 90 mL) at 4 °C for 24 h. The gradient alcohol dehydration method was used to soak them for 30 min in 70%, 80%, 90%, 95%, and 95% anhydrous ethanol, respectively. After the xylene had turned transparent twice, every 10 min. We put the sweetpotato root in the xylene and poured the melted paraffin on the precooled xylene. After the paraffin wax solidified, the material was placed in a wax melting furnace at 35–37 °C for 1–2 days. The transfer material was placed in a thermostat at 56–60 °C for 2–5 h. After the paraffin melted, the pure wax was replaced; the wax was changed repeatedly 2–3 times, and the xylene in the tissue was removed. We selected a suitable size for the embedded bottom mold, added the paraffin, put the tissue into the paraffin, and covered the disposable embedding box with a small amount of paraffin, which naturally cooled and solidified. Before slicing, we put the tissue in the refrigerator to refrigerate for 30 min, adjusted the thickness of the slice to 20–30 μm, and sliced it. First, we baked the slice at 65 °C for 30 min, then thoroughly dissolved the paraffin wax with xylene, usually dewaxing 2–3 times with xylene every 10 min. After dewaxing, the paraffin sections were sealed with resin, and the paraffin sections were observed and photographed under Leica biomicroscope (Leica-microsystems, Wetzlar, Germany).

#### 2.3.5. Expression Analysis of Saccharification-Related Genes in Sweetpotato

Total RNAs of all samples were extracted using the polysaccharide polyphenol plant RNA Isolation Kit (Huayueyang Biotechnology Co., Ltd., Beijing, China) following the manufacturer’s protocol. RNA integrity and concentration were verified, respectively, by 1.2% formaldehyde denaturing agarose gel electrophoresis and a Nanodrop spectrophotometer 1000 (ThermoFisher Scientifc Inc., Waltham, MA, USA). The cDNA was synthesized using a ReverTra Ace^®^ qPCR RT Master Mix with a gDNA Remover (TOYOBO, Osaka, Japan) kit. The sequences of saccharification-related genes were all from NCBI. qRT-PCR primers were designed using primer 3 plus online software (https://www.primer3plus.com/index.html, accessed on 5 December 2019) (Appendix A). *IbARF* was used as the internal control gene [20]. The mRNA levels were quantified by qRT-PCR amplification using a Quant Studio 6 system (Thermo Fisher Scientific Inc., Waltham, MA, USA) in a total volume of 20 μL, containing 10 μL of SYBR^®^ Green Realtime PCR Master Mix (TOYOBO, Osaka, Japan), 2 μL of cDNA templates, 1 μL of forward and reverse primer (10 mM), and 7 μL of ddH_2_0. The reaction program follows Kou’s method [21]. Gene transcript levels were calculated by the 2^−ΔΔCT^ method [22] and each sample was performed in triplicate.

### 2.4. Data Analysis

Three biological replicates were set for each indicator measurement. SPSS 22.0 software (International Business Machines Corporation, Amonk, NY, USA) was used for statistical analysis of data. Single-factor analysis of variance was used for each index among the three sweetpotato varieties, and the independent sample *t*-test was used between the two growth stages. We used OriginPro 2020 software (OriginLab corporation, Northampton, MA, USA) for graphics and Adobe Photoshop CS5 (Adobe Systems Incorporated, San Jose, CA, USA) for layout.

## 3. Results and Discussion

### 3.1. Analysis of Dry Matter Content, Starch, and Amylose Content

Z13, as a sweetpotato variety for both table use and processing [23], had the highest dry matter, starch, and amylose content. Y25 is a baked sweetpotato variety [24], and its dry matter content, starch content, and amylose content were low in the two growth stages. The two varieties were highly saccharified, and there was no significant difference in dry matter content, starch content, and amylose content between the two growth stages. X27 is a low saccharified variety [25], and its dry matter content and amylose content were in the middle level, but the starch content was high. In addition, the amylose content in 130 d growth period was significantly lower than that in 90 d growth period (Figure 1). There were some differences in dry matter content, starch content, and amylose content in the two growth stages of the three sweetpotato varieties. Starch and amylose content affected the saccharification efficiency of sweetpotato [26]. The content of amylose was significantly different under different temperatures and growth stages [27]. The results of this study also found that there were significant differences in amylose content of X27 in the two growth stages, but no significant differences in amylose content of Y25 and Z13 in the two growth stages, which may be related to the variety [28].

### 3.2. Analysis of Starch Content and Decomposition Rate

After being steamed, the starch content of the two high saccharified varieties Y25 and Z13 decreased significantly, indicating that starch was degraded. The starch degradation of the low saccharified variety X27 was less, and there was no significant change at both growth stages (Figure 2A,B). The starch degradation rates of Y25 and Z13 in both growth stages were more than 30%, while the starch degradation rate of X27 was less than 10% (Figure 2C). The starch decomposition rate of high sugar varieties (Y25 and Z13) was significant. The decomposition of starch during sweetpotato ripening is the direct source of soluble sugar. The more starch is decomposed, the more soluble sugar content is produced [16].

### 3.3. Analysis of Total Amylase Activity 

During the 90 d growth period, the total amylase activities of Y25 and Z13 were higher than those of X27 under different temperature treatments, while the total amylase of X27 was kept at a low level, showing a trend that increased at first and then decreased with temperature (Figure 3A). During the 130 d growth period, the total amylase activity of the three sweetpotato varieties increased at first and then decreased. However, the total amylase activity of Z13 and Y25 reached the highest point at 80 °C and 60 °C, respectively, and it was significantly higher than that of X27 (Figure 3B). Due to the high efficiency of amylase, a small increase in activity will exponentially increase the efficiency of amylase in decomposing starch [29]. Although there was no significant difference in amylase activity among the three sweetpotato varieties after partial temperature treatment for 130 d, the amylase activity of Y25 and Z13 was higher than that of X27. In addition, the stability of amylase at high temperature further determines the amount of maltose produced during sweetpotato ripening. The β-amylase activity isolated from the Japanese sweetpotato variety “QuickSweet” can still maintain high activity at 80 °C, while the β-amylase activity of the other sweetpotato variety “Benianuna” is seriously inhibited [30], indicating that the activity of amylase plays an important role in the decomposition of starch.

### 3.4. Analysis of Maltose, Sucrose, Fructose, Glucose, Soluble Sugar Contents, and Physiological Sweetness

With the treatment of different temperatures, the contents of sucrose, glucose, and fructose changed, but the regularity was not strong (Figure 4B–D,H,I,J), while the content of maltose increased with the increase in temperature in the two growth stages, and the content of maltose was the highest in the four kinds of sugars after ripening. The maltose content of Z13 and Y25 after ripening was significantly higher than that of X27 in both stages (Figure 4A,G). Soluble sugar and physiological sweetness showed the same increasing trend with the regular increase in maltose (Figure 4E,F,K,L), indicating that the final sweetness of sweetpotatoes depends on the content of maltose. The decomposition of starch to produce a large amount of maltose increased the physiological sweetness of sweetpotatoes after ripening. Related studies have also confirmed that the content of maltose determines the sweetness of sweetpotatoes [9,10]. During the curing process, the root tuber mainly produced a large amount of maltose due to the hydrolysis of starch catalyzed by amylase, while other soluble sugars barely changed. Therefore, the soluble sugars of cooked sweetpotato mainly contains four kinds of soluble sugars, namely maltose, sucrose, fructose, and glucose [31], which were consistent with the types of sugars measured in this study.

### 3.5. Changes of Starch Granules in Sweetpotato Storage Root during Heating

The changes of starch granules in cells during heating were observed by paraffin sections. The results showed that starch granules were mainly arranged and accumulated around the cell wall, which were spherical or hemispherical in size. The observed morphology of starch is consistent with relevant reports [32]. There was no significant change in the number and size of starch grains after treatment at CK, 50 °C, 60 °C, and 70 °C. After treatment at 80 °C, the number of starch grains began to decrease and flocs began to appear around the cell wall, indicating that starch grains began to gelatinize and decompose. After steaming, the cells decomposed, the flocs were obvious, and there were more fragments of starch grains and less intact starch grains, indicating that the starch grains were gelatinized and decomposed. Under different growth conditions, the root cell structure of sweetpotato tuber in the 90 d and 130 d growth period was highly consistent under different temperature treatments, and the starch grains began to decompose after 80 °C. At the same time, it was found that there were still more undecomposed starch grains in X27 after steaming but less undecomposed starch grains in Y25 and Z13. The degree of decomposition of starch grains in the cells of Y25 and Z13 was more perfect than that of X27 (Figure 5).

### 3.6. Analysis of Starch Gelatinization Characteristics

The starch gelatinization characteristics of three sweetpotato varieties, X27, Y25, and Z13, were analyzed. The results showed that the starch gelatinization temperature of Y25 and Z13 was significantly lower than that of X27 in the two growth stages. Related studies showed that there was a significant negative correlation between sweetpotato gelatinization temperature and maltose content. The lower the starch gelatinization temperature, the greater the maltose content [33]. This may be an important reason why Y25 and Z13 produce more maltose than X27. In addition, the gelatinization temperature of starch was also related to the growing soil temperature [34]. It was found that the gelatinization temperature of the three sweetpotato varieties in 130 days was significantly lower than that in 90 days (Figure 6C). However, the peak viscosity, final viscosity, trough viscosity, breakdown, and setback had no significant regularity in the two growth stages (Figure 6D–H). This phenomenon may be caused by the changes in the structure and properties of starch due to the difference between the two growth conditions [35,36].

### 3.7. Analysis of Sensory Evaluation, Physiological Sweetness, and Electronic Tongue

The sensory evaluation scores of Y25 and Z13 at two growth stages were significantly higher than that of X27, because the soluble sugar content of Xushu 27 was significantly lower than that of Y25 and Z13 (Figure 7A). According to the sweetness values of different sugar components, the physiological sweetness of ripe sweetpotato was calculated [37]. Although the sweetness of sucrose, glucose, and fructose was higher than that of maltose, the maltose content of Z13 and Y25 was significantly higher than that of X27 (Figure 7B), resulting in a significantly higher physiological sweetness than X27. Through the analysis of the taste composition of the three sweetpotato varieties, it was found that the electronic tongue patterns of the three sweetpotato varieties were quite different in the two growth periods, in particular, the electronic tongue map of Xushu 27 was more extreme (Figure 7C,D). 

### 3.8. Analysis of Expression of Glycation Related Genes

The expression of sugar metabolism-related genes in sweetpotato roots treated at different temperatures in two growth stages was determined by RT-qPCR. The results showed that the relative expression of glycosylation-related genes in Y25 roots in CK was significantly higher than that in the other two varieties (Z13 and X27) (Figure 8). During the 130-day growth period, many genes were significantly upregulated after X27 was treated at 50 °C, but this phenomenon was not found in the other two highly saccharified varieties. The results showed that there were differences in sugar metabolism-related genes in different growth stages of sweetpotato. At the same time, some saccharification genes were upregulated or downregulated by high temperature, but there were differences among varieties. Among them, the study found that regulating the expression of the amylase gene can effectively provide high soluble sugar content [38]. The level of isoamylase expression also affects the structure and characteristics of starch [39].

## 4. Conclusions

In this paper, the physical and chemical properties of two high saccharified varieties Y25 and Z13 and one low saccharified variety X27 were analyzed. The results show that there were significant differences in dry matter content, starch, and amylose content among the three varieties. Thus, affecting the gelatinization characteristics of starch, especially the gelatinization temperature, resulted in differences in maltose production, and finally, affected the sweetness and sensory evaluation of sweetpotato after ripening. The results provide an important reference basis for understanding the saccharification characteristics of different sweetpotato varieties and provide guidance for the selection of sweetpotato parents. In addition, a thorough understanding of the saccharification characteristics of sweetpotato varieties is of great importance to the promotion of varieties and full industrial utilization.

## Figures and Tables

**Figure 1 foods-12-03785-f001:**
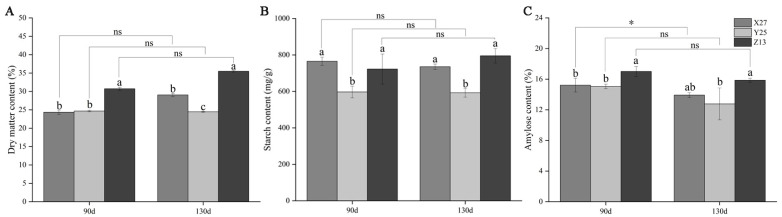
Changes of dry rate (**A**), starch (**B**), and amylopectin (**C**) content of three sweetpotato varieties X27, Y25, and Z13 at two growth stages. One-way ANOVA was used among varieties, and lowercase letters indicate that there were significant differences at *p* < 0.05. *t*-test analysis was used during childbearing period; * indicates that there was a significant difference at *p* < 0.05, while ns indicates that there was no significant difference.

**Figure 2 foods-12-03785-f002:**
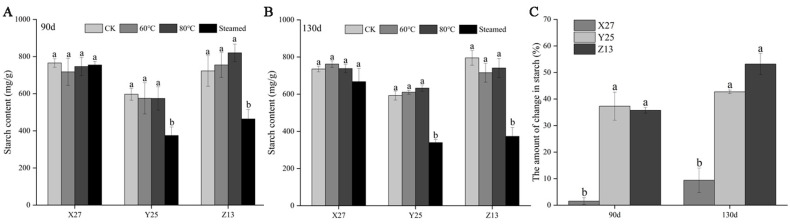
Starch decomposition of X27, Y25, and Z13 during heating. (**A**) Changes of starch content in different treatments of three sweetpotato varieties during the 90 d growth period. (**B**) Changes of starch content in different treatments of three sweetpotato varieties during the 130 d growth period. (**C**) Reduction in starch content of three sweetpotato varieties after ripening in two growth stages. One-way ANOVA was used between different treatments, and lowercase letters indicate that there were significant differences at *p* < 0.05.

**Figure 3 foods-12-03785-f003:**
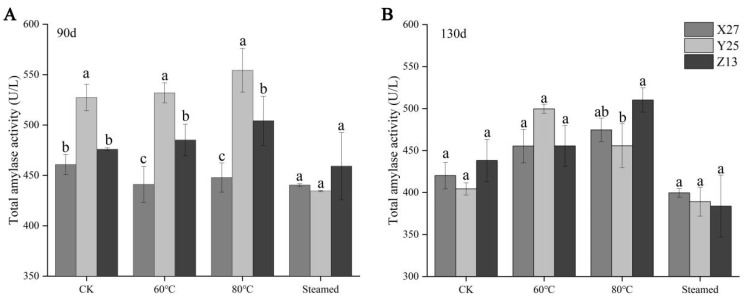
Changes of total amylase activity in X27, Y25, and Z13 under different temperature treatments. (**A**) Changes of total amylase activity in different treatments of three sweetpotato varieties during the 90 d growth period. (**B**) Changes of total amylase activity in different treatments of three sweetpotato varieties during the 130 d growth period. One-way ANOVA was used between different treatments, and lowercase letters indicate that there were significant differences at *p* < 0.05.

**Figure 4 foods-12-03785-f004:**
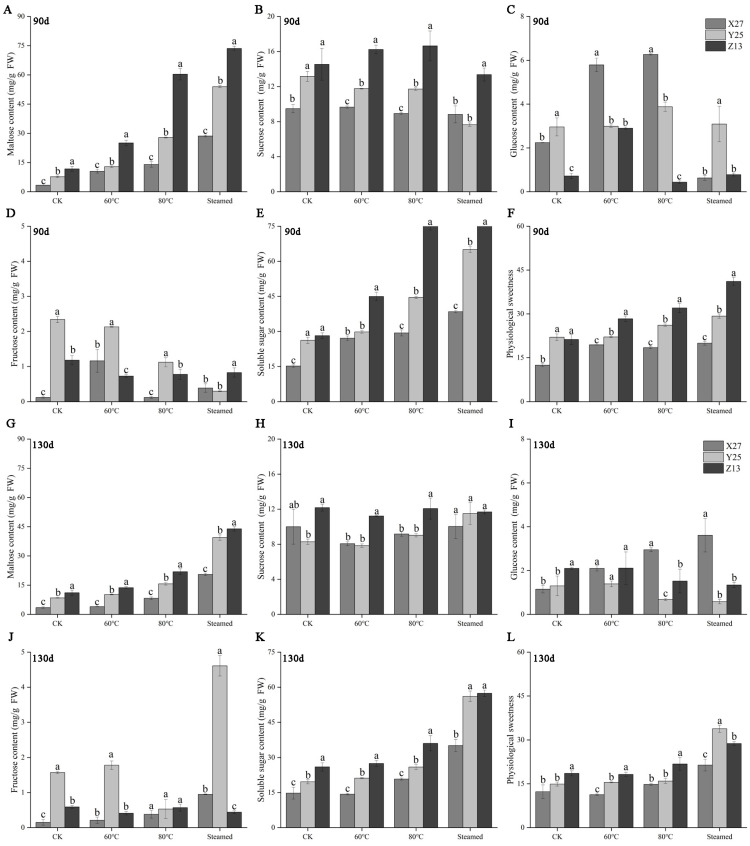
Changes of maltose, sucrose, glucose, fructose, soluble sugar content, and physiological sweetness of X27, Y25, and Z13 under different temperature treatments. Changes of maltose (**A**), sucrose (**B**), glucose (**C**), fructose (**D**), soluble sugar content (**E**), and physiological sweetness (**F**) in different treatments of three sweetpotato varieties during the 90 d growth period. Changes of maltose (**G**), sucrose (**H**), glucose (**I**), fructose (**J**), soluble sugar content (**K**), and physiological sweetness (**L**) in different treatments of three sweetpotato varieties during the 130 d growth period. One-way ANOVA was used between different treatments, and lowercase letters indicate that there were significant differences at *p* < 0.05.

**Figure 5 foods-12-03785-f005:**
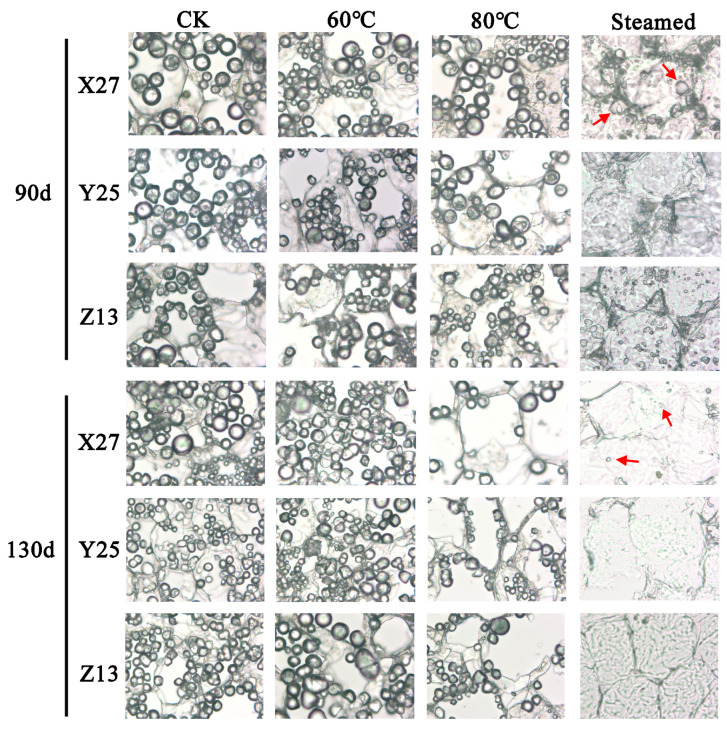
Paraffin sections of storage roots of X27, Y25, and Z13 treated at different temperatures. The microscope is magnified 400 times. The red arrows are undecomposed starch granules.

**Figure 6 foods-12-03785-f006:**
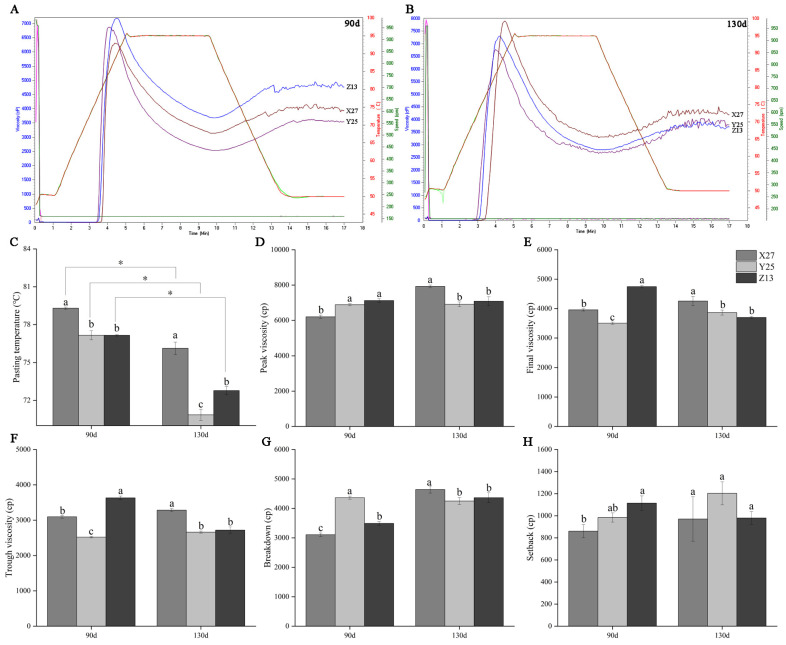
Analysis of starch gelatinization characteristics of X27, Y25, and Z13. (**A**,**B**) show the RVA curves of 3 sweet potato varieties on 90 d and 130 d, respectively. The blue, red, and green axes in A and B represent viscosity, temperature, and speed, respectively. (**C**–**H**) shows pasting temperature, peak viscosity, final viscosity, tough viscosity, breakdown and setback of three sweetpotato varieties at 90 d and 130 d, respectively. One-way ANOVA was used among varieties, and lowercase letters indicate that there were significant differences at *p* < 0.05. *t*-test analysis was used during childbearing period, and * indicate that there was a significant difference at *p* < 0.05.

**Figure 7 foods-12-03785-f007:**
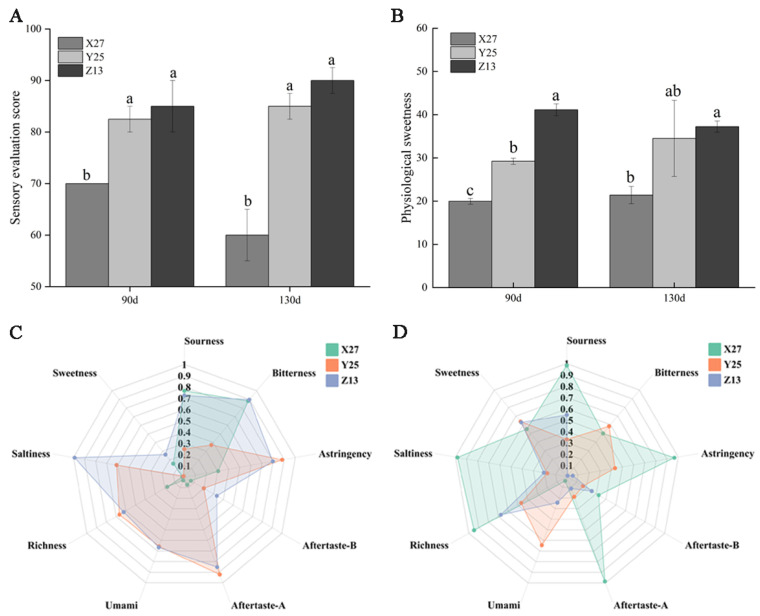
Sensory evaluation score, physiological sweetness, and electronic tongue analysis of X27, Y25, and Z13. One-way ANOVA was used among varieties, and lowercase letters indicate that there were significant differences at *p* < 0.05. (**A**,**B**) show the scores of sensory evaluation and physiological sweetness analysis respectively (**C**,**D**) show the electronic tongue maps of the three varieties at 90 d and 130 d, respectively.

**Figure 8 foods-12-03785-f008:**
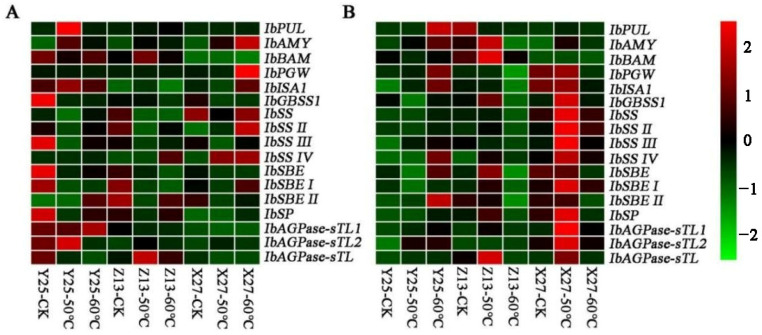
Analysis of glycosylation-related gene expression in X27, Y25, and Z13. (**A**,**B**) show the expression patterns of glycosylation-related genes for 90 days and 130 days, respectively.

## Data Availability

The data used to support the findings of this study can be made available by the corresponding author upon request.

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
