# Peer review of "Comparative Analysis of Saccharification Characteristics of Different Type Sweetpotato Cultivars"

_foods, 2023, doi:10.3390/foods12203785_

Round 1

Reviewer 1 Report

The paper entitled "Comparative analysis of saccharification characteristics of three sweetpotato cultivars "describes the physical and chemical properties of two high saccharified varieties 314 Y25 and Z13 and one low saccharified variety X27. Overall paper is well written however needs improvement in term of English language. In introduction part, new references can be added instead of old ones. Discussion part needs some more comprehensive interpretation of results. In conclusion author can provide the future prospects of this research work.

Author Response

According to the reviewer's suggestion, we have replaced part of the old literature, but some important references are still retained. In addition, the whole language of the paper has been modified and some prospect content has been added.

Reviewer 2 Report

Sweetpotato is one of the important sources of carbohydrate and starch that usually process in industry. The studied about saccharification of several sweetpotato cultivars is important to know the characteristics of sweetpotato so industry cloud make choice of sweetpotato as raw material. The author's study would be excellent for further utilization of the exploration saccharification. However, it requires several improvements.

Title:

The title was not clear enough to define the focused study,  the title mention comparation in 3 different cultivars of sweetpotato but result show not only the comparation of 3 cultivars but also 3 different heat process of the sweetpotato.

Abstract:

- more elaborate the background of the research

- Line 16-18 “The saccharification characteristics of sweetpotato determine the edible quality, however few related studies are available. A comparison study of two high sac charification varieties (Y25 and Z13) and one low saccharification variety (X27) were selected to analyze their storage roots physical and chemical properties” is confusing about that the saccharification group is determined at the beginning while in this study these characteristics is about to investigate.

- Provide the complete important result in abstract

Keywords:

- It would be better to use 3-5 words that different from manuscript title to enhance the discoverability

Introduction:

- “The table use sweetpotato needs high saccharification efficiency, producing more soluble sugar and improving taste” this sentence is unclear please rewrite the sentence. 

- Please justify why Y25 Z13 and X27 cultivars is use as samples of this research  

- some information of the example of product form industry that need low saccharification and high saccharification shall be provided. 

- The second paragraph is very long. 

- Please rewrite the objective of this study briefly in the last paragraph

Material and Method

- Add detailed information about the producer, city, and the country of origin for every instruments, materials, and software.

- Please explain the meaning of “CK” in line 94 

- line 94: add more information why the temperature for treatment was chosen

- line 102: freeze-drying is a tool for drying not for determined the dry matter content

- Line 107: washing is not one of the methods of extraction

- please provide more information of each analysis in section 2.3.3 and divided into several deferent section 

- line 127: mention the software used for the analysis.  

Result and Discussions

- For the title of section in result and discussion should be written concisely, more effective and indicates what section will be discussed in the paragraph instead of the result.

- Section 3.1 please compare with other study.  

- section 3.2 please elaborate more the explanation of this section and compare with other study. 

- what the meaning of degree of decomposition and how to determine it? 

for all the discussion, please elaborate more 

Overall

- Several units was not written properly, it should be written separately from the value except “%”

Author Response

Sweetpotato is one of the important sources of carbohydrate and starch that usually process in industry. The studied about saccharification of several sweetpotato cultivars is important to know the characteristics of sweetpotato so industry cloud make choice of sweetpotato as raw material. The author's study would be excellent for further utilization of the exploration saccharification. However, it requires several improvements.

  1. Title:

The title was not clear enough to define the focused study, the title mention comparation in 3 different cultivars of sweetpotato but result show not only the comparation of 3 cultivars but also 3 different heat process of the sweetpotato.

Because the saccharification characteristics of sweetpotatoes include changes in sugar content during the ripening process, different temperature treatments are also an important aspect of examining the saccharification characteristics of sweetpotatoes. Therefore, we believe that the title is not a problem.

  1. Abstract:

- more elaborate the background of the research

We have supplemented the research background in the abstract.

- Line 16-18 “The saccharification characteristics of sweetpotato determine the edible quality, however few related studies are available. A comparison study of two high sac charification varieties (Y25 and Z13) and one low saccharification variety (X27) were selected to analyze their storage roots physical and chemical properties” is confusing about that the saccharification group is determined at the beginning while in this study these characteristics is about to investigate.

We have found significant differences in sugar content and taste among the three sweet potato varieties Y25, Z13, and X27 during daily processing. Y25 and Z13 have high sweetness, while X27 has low sweetness, so we believe that Y25 and Z13 are high saccharification varieties and X27 is low saccharification varieties. However, this article provides a more detailed analysis of the differences in specific saccharification characteristics

- Provide the complete important result in abstract

We have supplemented the specific results of the study in the abstract.

  1. Keywords:

- It would be better to use 3-5 words that different from manuscript title to enhance the discoverability

We have made modifications to the keywords based on the reviewer's comments and have highlighted them in red in the text.

  1. Introduction:

- “The table use sweetpotato needs high saccharification efficiency, producing more soluble sugar and improving taste” this sentence is unclear please rewrite the sentence.

According to the reviewer's comments, modifications have been made and highlighted in red in the text.

- Please justify why Y25 Z13 and X27 cultivars is use as samples of this research.

We have added the reasons for choosing these three varieties in the materials section and marked them in red.

-some information of the example of product form industry that need low saccharification and high saccharification shall be provided.

According to the reviewer's suggestion, relevant content has been added in a few sections and highlighted in red.

- The second paragraph is very long.

We have rephrased the second paragraph of the introduction section.

- Please rewrite the objective of this study briefly in the last paragraph

We have briefly explained the purpose of the study in the last paragraph of the introduction section.

  1. Material and Method

- Add detailed information about the producer, city, and the country of origin for every instruments, materials, and software.

We have added information on manufacturers and cities of instruments, software, etc. to the article.

- Please explain the meaning of “CK” in line 94

We have added and marked in the article.

-line 94: add more information why the temperature for treatment was chosen

We have added information about temperature selection to the text.

- line 102: freeze-drying is a tool for drying not for determined the dry matter content

- Line 107: washing is not one of the methods of extraction

The water washing method in this paper refers to the extraction of starch by water washing and precipitation, and is not a simple water washing. 

- please provide more information of each analysis in section 2.3.3 and divided into several deferent section

According to the reviewer's comments, the content of section 2.3.3 has been supplemented and modified and marked in red.

-line 127: mention the software used for the analysis.

The analysis software has been marked in the paper.

  1. Result and Discussions

- For the title of section in result and discussion should be written concisely, more effective and indicates what section will be discussed in the paragraph instead of the result.

We have modified this part according to the reviewer's suggestion.

- Section 3.1 please compare with other study.

We have added to the relevant content.

- section 3.2 please elaborate more the explanation of this section and compare with other study.

We have added to the relevant content.

- what the meaning of degree of decomposition and how to determine it?

In this paper, the degree of starch decomposition can be measured by the difference between starch content of fresh potato and starch content after curing. The greater the difference, the higher the degree of decomposition.

- for all the discussion, please elaborate more

We re-added the discussion section.

Overall- Several units was not written properly, it should be written separately from the value except “%”

According to the reviewer's comments, modifications have been made and highlighted in red in the text.

Reviewer 3 Report

Comments

Results and discussion

Please heading 3.1 re-written it, use a single phrase

Line 174 please refers data to a proper Figure, similar comment to lines 175 -178.

Figure 1, please re-organize footnote in the same page

Please  re-written heading 3.2

Please re-written heading 3.3

Please re-written heading 3.4

Please re-written headings 3.5 to 3.8

Author Response

  1. Results and discussion

Please heading 3.1 re-written it, use a single phrase

We have been rewritten using a single phrase.

Line 174 please refers data to a proper Figure, similar comment to lines 175 -178.

We have made appropriate adjustments to the relevant content.

Figure 1, please re-organize footnote in the same page

Appropriate modifications have been made.

Please re-written heading 3.2,3.3,3.4,3.5-3.8

Some titles have been rewritten according to the reviewer's suggestion.
